# Productivity growth, economies of scale and scope in the water and sewerage industry: The Chilean case

**Maria Molinos-Senante** [1,2]*, **Alexandros Maziotis** [1,3]

**1** Departamento de Ingeniería Hidráulica y Ambiental, Pontificia Universidad Católica de Chile, Santiago, Chile, **2** Centro de Desarrollo Urbano Sustentable CONICYT/FONDAP/ 15110020, Santiago, Chile, **3** Department of Business, New York College, Athens, Greece

\* mmolinos@uc.cl

## Abstract

Evaluating the performance and analyzing the cost drivers of water utilities is of great interest for water regulators and water sector managers. This study uses a quadratic cost function to investigate the existence of economies of scale and scope in the Chilean water and sewerage industry over the period 2010–2017. We also estimate and decompose productivity growth into technical change and scale efficiency change. Technical change is further broken into pure, non-neutral and scale-augmenting technical change. The results indicate that cost savings can be achieved by increases in the scale of production and the separation of water and sewerage services. Productivity progressed favorably throughout the whole period at an annual rate of 8.4%, which was attributed to the scale effect, the adoption of new technologies and a good allocation of resources. Some policy implications are finally discussed based on our findings.

## 1. Introduction

Drinking water services were traditionally provided to people by public utilities [1]. Over the previous decades several countries worldwide have experienced either a full privatization of water services such as in England and Wales, and Chile or more private participation of water services in the form of Public-Private Partnerships (PPPs) such as in Spain, France and Portugal [2, 3]. In particular, the privatization of the water utilities as natural monopolies requires the establishment of regulator to guarantee the protection of customers and environment, and the financial sustainability of the water companies. The concept of natural monopoly in the water industry is associated with the optimal industry configuration (economies of scale and scope) and efficiency and productivity.

Economies of scale show how output change when there is an increase in inputs [4]. Increasing economies of scale exist when an increase in the production of more output (e.g., through mergers and acquisitions) leads to a lower increase in production costs [5]. Economies of scope may exist due to cost savings from the joint production of different types of services (e.g., water and sewerage services) [6]. Efficiency and productivity change measure the ability

committee. Data are available from the Superintendencia de Servicios Sanitarios Institutional Data Access. Data requests may be made to rfarias@siss.gob.cl or transparency process (https://www.portaltransparencia.cl/PortalPdT/directorio-de-organismos-regulados/?org=AM011).

**Funding:** The authors received no specific funding for this work.

**Competing interests:** The authors have declared that no competing interests exit.

of the firm (water company) to minimize its costs for a given level of outputs and show how less efficient firms improve (or deteriorate) performance over the most efficient firms while the latter are improving (or are deteriorating) their performance over time.

The concept of economies of scale and scope in the water industry had been researched by several authors in different developed countries such as England and Wales, Portugal and Italy [4, 6–12] along with the concept of efficiency and productivity change [13–15]. However, a recent literature review by Cetrulo et al. [16] demonstrated that these concepts have received limited research in developing countries mainly due to low availability of data. Our study fills this gap in literature by estimating economies of scale and scope and total factor productivity growth (TFP) in the Chilean water industry. The concept of economies of scale is of great importance for the water regulator and regulated companies for the following reasons. An industry that operates under increasing economies of scale suggests that further adjustments in companies' scale of operations could lead to lower costs. This means that the regulator could promote mergers among companies as cost savings could be achieved if companies become bigger. In contrast, if the industry operates under decreasing economies of scale, then further increases in the size of the companies could not be justified from a cost perspective. This implies that the regulator should not promote policies that merge companies as overall costs would not go down. The concept of economies of scope also receives considerable attention by water regulator and regulated companies. The existence of positive economies of scope between water and sewerage services suggests that the joint provision of these services brings substantial cost savings. This means that water and sewerage companies should remain integrated, and the regulator should not promote policies that could unbundle these services. Breaking up the water and sewerage companies into smaller companies that focus on the provision of water services only and on the supply of sewerage services only might not lower production costs. By contrast, negative economies of scope imply that cost savings could be achieved if the water and sewerage services are provided by separate utilities. Understanding therefore the cost structure of the industry could help policy makers to deliver the water and wastewater services in an efficient and affordable manner. Therefore, the regulator and regulated companies have a significant role to play. For instance, the regulator should protect the customers from the monopoly abuse of the water companies. Moreover, it should monitor the performance of the companies by ensuring that any costs savings are passed to the customers in terms of lower prices. Finally, the regulator should ensure that sufficient finance is available to the water companies to operate and upgrade the network so that enough water is available for present and future generations.

Like England and Wales, the Chilean water industry is fully privatized and its privatization took place during the years 1998–2004 [17]. (Two forms of companies were formed namely: i) full private water companies (FPWCs) where the full ownership and operation of infrastructure is undertaken by private consortiums for an infinite time period, and ii) concessionary water companies (CWCs) where the water services are provided by private consortiums for a limited time period (e.g., 30 years) through a concessionary contract [18]. A small proportion of customers is provided by a public water company as well [19]. The monopolistic nature of the Chilean water companies required the establishment of a national regulator, the Superintendencia de Servicios Sanitarios (SISS). Its duties are to ensure the financial sustainability of the sector, set tariffs and keep customers' bills at an affordable level, enhance economic efficiency and water conservation [20]. Tariffs are set based on the definition of a hypothetical efficient water company (a "model company") [21]. They need to cover operating and capital expenditure, taxes and guarantee a fair return to capital invested. Water companies need to ensure that sufficient water is available for people which might be difficult in the framework of climate change. To achieve this objective, water companies should buy water rights from other

water users such as agriculture or mining at market price. Given the model applied in Chile to set water tariffs, the purchase of water rights by the water companies might impact on water tariffs. However, detailed studies by Donoso [22] and SISS [23] evidenced that the scarcity value of water, i.e., the price of water rights, is not reflected in the Chilean water tariffs.

Studies by Molinos-Senante and Sala-Garrido [17] and Sala-Garrido et al. [19, 24] used non-parametric techniques to study the productivity change of FPWCs and CWCs, i.e., the public water company was not integrated in the assessment. Studies by Molinos-Senante et al. [18, 25] used parametric techniques to study the performance of the private Chilean water industry. The main disadvantage of non-parametric techniques over parametric techniques lies in their inability to separate noise from inefficiency. However, Molinos-Senante et al. [18, 25] did not study the concept of economies of scale and scope between the provision of water and wastewater services. Our study aims to fills this gap in literature.

To overcome the above limitations, our study uses parametric techniques to analyze the performance of Chilean water companies, both private (FPWCs and CWCs) and public one. In doing so, we estimate a quadratic cost function that allows us to quantify the impact of economies of scale and scope between water and sewerage services and further estimate and decompose TFP in several factors of interest. A detailed justification of the use of the quadratic cost function is provided in the "methodology" section. This study contributes to the existing literature in the following ways. First, for the first time a quadratic cost function is used to estimate economies of scale and scope and TFP growth in the Chilean water industry. Second, the use of quadratic cost function allows us to quantify the cost savings from the joint or separate production of water and sewerage services. Third, the use of this cost function allows us to estimate and decompose productivity growth into technical change and scale change. A further decomposition of technical change into pure effect, non-neutral and scale augmenting change is provided. Fourth, we are able to compare productivity performance by ownership type and thus, we provide more evidence on the ownership-privatization nexus of water utilities. To the best of our knowledge, there are not any studies in the Chilean water industry that use a quadratic cost function to estimate economies of scale and scope and TFP of private and public water utilities. This approach focuses on several Chilean water companies that provide water and sewerage services over the years 2010–17. Finally, some policy implications are discussed as well.

The paper unfolds as follows. The next section presents the methodology used followed by a section on the data and variables used in this study. Section 4 analyzes the results, and the final section concludes.

## 2. Methodology

The estimation of economies of scale and scope and TFP assessment was carried out using the quadratic cost function which was selected for the following reasons. Firstly, we wanted to estimate the productivity growth of the water companies using discrete data and this was done with the use of a quadratic cost function. Secondly, it is easy to apply and estimate a quadratic cost function and interpret its results [26]. Moreover, it allowed us to quantify in monetary terms the savings from the joint or separate production of multi-output services. Thirdly, the quadratic cost function has been frequently used in the literature to analyze the cost structure of regulated industries, i.e., the estimation of economies of scale and scope [27, 28]. However, it has hardly been applied to analyze productivity change of regulated sectors [29, 30]. Moreover, while several studies have used a translog cost specification to estimate productivity change in water industry [18, 31], to the best of our knowledge, the quadratic cost function to estimate the productivity change on the water sector has been hardly researched. Furthermore, as Triebs et al. [26], Molinos-Senante and Maziotis [12, 32] noted that there are several advantages for

choosing a quadratic cost function over other functional forms such as a translog, composite or generalized composite which have also been used in other studies [9, 10, 33, 34]. Compared to translog function, a quadratic cost function takes into account zero values. Other functional forms e.g., composite or generalized composite are difficult to estimate and converge and in other cases their coefficients do not have a direct economic meaning [26]. Limitations of the quadratic cost function involve the imposition of homogeneity in input prices as a regularity condition which may sacrifice flexibility [35]. However, this issue can be dealt with by normalizing cost and input prices by one of the input prices prior to estimation [29, 32].

We defined economies of scale and scope for a quadratic cost function following the concept of Baumol et al. [36]. Assuming that a water company produces a vector of outputs $y_n$ where $y_n = (1, \ldots, N)$, the economies of scale can be defined as follows:

$$ESCALE = \frac{C(y)}{y \times \nabla C(y)} = \frac{C(y)}{\sum_{n=1}^{N} y_n C_n(y)} \qquad (1)$$

where $C(y)$ denotes the cost of producing the vector of outputs and $C_n(y)$ presents the marginal cost of producing the output. Economies of scale with a value higher than one indicates that the water company operates under increasing economies of scale which means that there will be a less proportionate increase in average costs if outputs increase as well [5]. A water company operates under diseconomies of scale if $ESCALE$ takes a value less than one.

Economies of scope between $K$ and $N - K$ products within the entire output vector $N$ were defined as follows:

$$C(y_N) < C(y_K) + C(y_{N-K}) \qquad (2a)$$

where in Eq (2a) the cost of producing the entire output vector $N$ is less than the cost of producing separately the product $K$ and the $N - K$ products. In percentage term, Eq (2a) can be expressed as follows [37]:

$$ESCOPE = \frac{C(y_K) + C(y_{N-K}) - C(y_N)}{C(y_N)} \qquad (2b)$$

Economies of scope ($ESCOPE$) may take a positive, negative or zero value. A positive value means that there are economies of scope between $K$ and $N - K$ products which suggests that the water company achieves cost savings by jointly producing these outputs than separately. A negative value means that there are diseconomies of scope between $K$ and $N - K$ outputs. This means that the water company benefits by producing these products separately than jointly. If economies of scope take a zero value, then it implies that the joint production cost of the entire product $N$ is the same as generating $K$ and $N - K$ separately.

Following past practice [12] we defined the following normalized quadratic cost function (NQCF) by normalizing total costs with one of the input prices [29]:

$$NQCF = \alpha_o + \sum_{i=1}^{n} \beta_i y_i + \sum_{i=1}^{m} \gamma_i w_i + \frac{1}{2} \sum_{i=1}^{n} \sum_{j=1}^{n} \beta_{ij} y_i y_j +$$
$$\frac{1}{2} \sum_{i=1}^{m-1} \sum_{j=1}^{m-1} \gamma_{ij} w_i w_j + \sum_{i=1}^{n-1} \sum_{j=1}^{m} \delta_{ij} y_i w_j + \varphi_1 t + \sum_{j=1}^{n} \lambda_j y_j t + \sum_{i=1}^{m-1} \pi_i w_i t + \qquad (3)$$
$$\frac{1}{2} \varphi_2 t^2 + \sum_{r=1}^{R} \rho_i \xi_i + \sum_{i=1}^{N} \psi_i D_i$$

where $NQCF$ denotes the normalized total cost of production, $w$ is a vector of normalized input prices where $m$ is the total number of prices, $y$ is a vector of outputs where $n$ is the total number of outputs, $t$ is a time trend, $\xi$ is a set of environmental variables which may influence

costs [12]. We also captured water companies´ heterogeneity by adding firm specific dummies $D_i$ [30]. As Kwoka [37] and Stone and Wester Consultants [7] noted we can link Eqs (2a and 2b) with Eq (3) as follows in the case of $n$ outputs:

$$ESCOPE = (n-1)a_0 - \sum_{i=1}\sum_{j>1}^{n}\beta_{ij}y_iy_j \tag{4}$$

where $a_0$ is the normalized total cost of the production process and cost complementarities between the two outputs are captured by their interaction term, $\beta_{ij}y_iy_j$. A positive value of Eq (4) involves positive economies of scope whereas a negative value denotes diseconomies of scope.

Following past practice [30], we applied Shephard's Lemma to derive the input demand equation which takes the following form:

$$x_i = \gamma_i + \pi_i t + \sum_{j=1}^{n-1}\gamma_{ij}w_j + \sum_{j=1}^{m}\delta_{ij}y_j \tag{5}$$

In Eqs (3) and (5) we added classic errors terms and estimated the equations simultaneously using Zellner's [38] iterated seemingly unrelated regression (SUR) technique [32]. We used this method because it increases the number of degree of freedom and does not add more parameters leading therefore, to more efficient estimates [26, 29]. After we normalized costs and input prices by one arbitrarily input price, we removed the corresponding input demand equation from estimation [12].

After estimating the quadratic cost function and its input demand equations we used Marti-nez-Budria et al.'s [29] approach to derive and decompose productivity growth ($T\dot{F}P$) between two time periods, $t_0$ and $t_1$ as follows:

$$T\dot{F}P = -\dot{D} + \dot{N}\left[1 - \frac{1}{2}\sum_{J=1}^{n}\left(\frac{C_{t_1}\dot{y}_{J_0}}{C_{t_0}y_{J_1}}\varepsilon_{C,y_{J_1}} + \varepsilon_{C,y_{J_0}}\right)\right] \tag{6}$$

where $\varepsilon_{C,y}$ is the cost elasticity of output and:

$$\dot{D} = \frac{1}{2C_{t_0}}\left[\left(\frac{\vartheta C}{\vartheta t}\right)_{t_1} - \left(\frac{\vartheta C}{\vartheta t}\right)_{t_0}\right](t_1 - t_0) = \left[\frac{C_{t_1}}{2C_{t_0}}\dot{T} + \frac{1}{2}\dot{T}\right](t_1 - t_0) \tag{7}$$

Where $\dot{T}$ denotes the technical change (TC) which was estimated as:

$$\dot{T} = \frac{1}{C}\frac{\vartheta C}{\vartheta T} \tag{8}$$

$\dot{N}$ is an index of change in outputs between the two periods:

$$\dot{N} = \frac{\sum_{j=1}^{n}\left(\frac{C_{t_1}y_{j_0}}{C_{t_0}y_{j_1}}\varepsilon_{C,y_1} + \varepsilon_{C,y_{j_0}}\right)\dot{y}_{J_0}}{\sum_{j=1}^{n}\left(\frac{C_{t_1}y_{j_0}}{C_{t_0}y_{j_1}}\varepsilon_{C,y_1} + \varepsilon_{C,y_{j_0}}\right)} \tag{9}$$

If the output vector changes proportionally and the water company operates under constant returns to scale, then there is no scale effect and technical change is the only determinant of productivity change. If increasing returns to scale exist, then the productivity index is larger than technical change index, whereas the presence of decreasing returns to scale implies that the productivity index is smaller than technical change index [39].

Moreover, technical change in Eq (8) was calculated as follows:

$$\frac{\vartheta C}{\vartheta t} = \phi_1 + \phi_2 t + \sum_{i=1}^{m-1} \pi_i w_i + \sum_{j=1}^{n} \lambda_j y \tag{10}$$

As noted by Baltagi and Griffin [40], a further decomposition of TC into three components is shown in Eqs (11–13):

Pure Technical Change (PTC):

$$PTC = \frac{1}{C}(\phi_1 + \phi_2 t) \tag{11}$$

Non-neutral technical change (NNTC):

$$NNTC = \frac{1}{C}\left(\sum_{i=1}^{m-1} \pi_i w_i\right) \tag{12}$$

Scale-augmenting technical change (SATC):

$$SATC = \frac{1}{C}\left(\sum_{j=1}^{n} \lambda_i y_i\right) \tag{13}$$

PTC shows how costs change over time due to the adoption of new technologies. NNTC suggests that changes in technology did not favor any of the inputs used. Consequently, NNTC implies that new technologies may have led to the use of more input than others e.g., more capital than operating costs. This may be attributed to changes in the prices of the inputs. SATC shows how costs change over time when new technologies may lead to changes in the mix of outputs.

## 3. Description of sample data

The data employed to carry out the empirical application was obtained from the website of the Chilean water regulator (Superintendencia de Servicios Sanitarios, SISS) and involves the period 2010–2017. Following past practice [15, 41, 42], we defined two outputs namely: i) volume of drinking water delivered measured in thousands of cubic metres per year and ii) number of customers receiving wastewater collection and treatment. Total costs were defined as the sum of operating expenditure of water and sewerage services and capital expenditure which consists of companies' investments to improve the water and sewerage network. Total costs were expressed in Chilean pesos per year and were deflated by the consumer price index taken from national statistics. The price of capital was calculated as the ratio of capital expenditure and the financial value of physical assets [43, 44]. The price of operating expenditure was approximated by the Chilean producer price index taken from national statistics [12, 44]. We used the price of operating expenditure to normalize costs and input prices.

Finally, as several studies noted [14, 18, 45–48] there might be several operating characteristics that could influence water companies' total costs and so they should be included in the analysis. Thus, our model incorporates the following variables: i) customer density measured as the number of customers divided by the network length; ii) non-revenue water denoted as the percentage of water that is produced but is not billed; iii) type of water resource, a categorical variable to capture that water is taken from surface, groundwater and mixed (both surface and groundwater) resources.

Table 1 shows the descriptive statistics used in the sample.

**Table 1. Descriptive statistics for the Chilean water and sewerage utilities over the years 2010–2017.**

| Variables | Unit of measurement | Mean | Std. Dev. | Minimum | Maximum |
|---|---|---|---|---|---|
| Total costs | $10^3$CLP / year* | 51159525 | 69354174 | 1265731 | 357261217 |
| Volumes of water delivered | $10^3$ m$^3$ / year | 54685 | 95299 | 1304 | 458025 |
| Customers receiving wastewater treatment | nr | 791161 | 1345086 | 17249 | 6451025 |
| Price for capital | CLP per year | 0.0662 | 0.0525 | 0.0073 | 0.2773 |
| Price for operating expenditure | Price index | 0.930 | 0.072 | 0.826 | 1.027 |
| Customer density | Customer / nr/ km | 58.81 | 13.90 | 19.41 | 87.21 |
| Non revenue water | % | 32.45 | 10.04 | 9.30 | 51.20 |
| Type of water resource | Categorical | 1.381 | 0.724 | 0.00 | 2.00 |

Observations: 160

Source: Own elaboration from Superintendencia de Servicios Sanitarios data.

*Operational and capital costs were adjusted to nominal CLP by the Chilean Consumer Price Index.

The exchange rate on February, 17th 2021 was USD 1 CLP 718 and EUR 1 869 CLP

## 4. Results and discussion

### 4.1 Normalized quadratic cost function

The results from the estimation of the NQCF and its input demand are reported in Table 2. Variables were subtracted from their average values and therefore, they allow for a direct

**Table 2. Estimated parameters of the normalized quadratic cost function.**

| Variables | Parameters | Coeff. | St.Error | T-stat | P-value |
|---|---|---|---|---|---|
| Constant | $a_0$ | 59169700 | 6814120 | **8.683** | 0.000 |
| Water delivered | $\beta_1$ | 292.0 | 86.4 | **3.379** | 0.000 |
| Wastewater treatment | $\beta_2$ | 47.1 | 5.6 | **8.370** | 0.000 |
| Capital price | $\gamma_1$ | 28037800 | 618959 | **45.298** | 0.000 |
| Time | $\varphi_1$ | 150111 | 194254.31 | 0.772 | 0.440 |
| Water delivered$^2$ | $\beta_{11}$ | -0.012 | 0.003 | **-4.091** | 0.000 |
| Wastewater treatment2 | $\beta_{22}$ | -0.001 | 0.000 | **-4.375** | 0.000 |
| Water delivered*water treatment | $\beta_{12}$ | 0.001 | 0.000 | **4.054** | 0.000 |
| Capital price$^2$ | $\gamma_{11}$ | 486.6 | 825.9 | 0.589 | 0.556 |
| Water delivered*capital price | $\delta_{11}$ | 243.8 | 59.7 | **4.081** | 0.000 |
| Wastewater treatment* capital price | $\delta_{21}$ | 327.8 | 9.6 | **34.265** | 0.000 |
| Water delivered*Time | $\lambda_1$ | 30.9 | 11.5 | **2.680** | 0.007 |
| Wastewater treatment* Time | $\lambda_2$ | -2.025 | 0.814 | **-2.488** | 0.013 |
| Capital price*time | $\pi_1$ | 522441 | 260081 | **2.009** | 0.045 |
| Time$^2$ | $\varphi_2$ | -184502 | 94414.14 | *-1.954* | 0.051 |
| Customer density | $\rho_1$ | 16611200 | 17992800 | 0.923 | 0.356 |
| Type of water resource | $\rho_2$ | 1389350 | 698000 | **1.990** | 0.047 |
| % non-revenue water | $\rho_3$ | 13979100 | 12330400 | 1.134 | 0.257 |
| Ownership | $\rho_4$ | -3870060 | 656118 | **-5.898** | 0.000 |
| Log-likelihood | | 44.610 | | | |
| Breuch pagan test | | $X^2 = 49.5$ | | | |

Dependent variable: C = normalized total cost of production

Bold coefficients are statistically significant from zero at the 5% level

Bold italic coefficients are statistically significant from zero at the 10% level

interpretation of the parameters [32, 49]. Water company-specific dummies are all statistically significant from zero and are not reported here due to spacious reasons. The estimated normalized constant (CLP 59,169,700), which can be interpreted as the estimate total cost, is close to the average un-normalized total cost (CLP 51,159,525). As expected, the first order coefficients of outputs and input price are all positive and statistically significant from zero. This is a property that is satisfied for all observations which means that the monotonicity condition is satisfied [29].

On average, the marginal cost of delivering an additional unit of water is CLP 292 (or € 0.350 m$^3$), whereas the provision of wastewater treatment to one more customer is CLP 47,000 (or € 56.5). Both the marginal cost of water delivered and customers receiving wastewater treatment decreased over time as shown by the negative sign of their quadratic terms, which are statistically significant. The sign of the interaction term between these two outputs is positive and statistically significant, which suggests the existence of diseconomies of scope. This means that on average the marginal cost of producing water delivered increases the marginal cost of wastewater treatment, and vice versa.

Derived demand for capital increases average total costs by CLP 28,037,800 per annum as shown by its positive and statistically significant coefficient. It appears that changes in technology increased the demand for capital and therefore, average total costs by CLP 522,441 per year. This can be seen by the positive and statistically significant coefficient of capital price and time. Moreover, time trend is captured by the parameter $t$, and is shown to be positive. This implies a trend rate of total cost increase of CLP 150,111 or 0.254% (calculated as the coefficient of time divided by constant term) per year. However, this result is not statistically significant.

The results from the operating characteristics suggest that the type of water resource and ownership had an influence on average total costs. In particular, the more groundwater and mixed water resources the water company has (than surface), the higher the total costs it may incur. This is attributed to the fact that abstracting water from this type of water resources might require high energy which could lead to higher operational costs. Moreover, as water availability decreases, the depth of wells increases and therefore, higher energy requirements might be needed to pump and treat water from groundwater and mixed water resources which could have an impact on operational costs. The negative sign on the variable ownership means that private companies may incur lower average costs than the public water company.

### 4.2 Economies of scale and scope

Table 3 reports the results from the estimation of economies of scale and scope for the average Chilean water company. It is found that on average the Chilean water and sewerage companies operated under increasing economies of scale. This result is statistically significant from zero. This means that a 1% proportional expansion of water delivered and wastewater treatment will lead to an increase in average total costs by 0.85%. Thus, any increases in the scale of operations of the average company, for instance, through mergers, could lead to lower costs. The

**Table 3. Cost elasticities, economies of scale and scope for the average Chilean water company.**

|  | Water delivered | Wastewater treatment |
|---|---|---|
| Cost elasticity | **0.238** | **0.613** |
| Economies of scale | **1.176** | |
| Economies of scope (%) | -43.4 | |
| Economies of scope (10$^3$ CLP/year) | -25,659,600 | |

Bold coefficients are statistically significant from zero at the 5% level

existence of increasing economies of scale in the Chilean water industry was also reported by previous studies [18, 50]. This finding suggests that if water companies want to increase the size of their operations by acquiring more customer base, it could lead to lower production costs. This means that neighbor companies could potentially merge to benefit from economies of scale. Currently, this option is not feasible because each water company has defined its concession area (i.e., operational territory) where it can provide water and sewerage services. However, this a policy that the regulator could potentially consider as it can be justified from a costs perspective.

By contrast, statistically significant negative economies of scope between water and wastewater services were found. This means that the separate production of volumes of water delivered and customers receiving sewage treatment would reduce costs by 43.4% on average. In monetary terms, this is equivalent, to a reduction in costs by $10^3$ CLP 25,659,600 per year or 30,791,520 €/year on average. To the best of our knowledge, we are not aware of any published research regarding economies of scope between water and sewerage services in Chile. Thus, we can compare our findings with previous studies on scope economies in the water industry beyond Chile. For instance, in England and Wales, Stone and Webster Consultants [7] found diseconomies of scope between water and sewerage services by specifying several cost functions (e.g., translog, quadratic). Bottasso et al. [34] also found diseconomies of scope between water and sewerage services in England and Wales by specifying a generalized composite cost function. Thus, our findings corroborate the results of the above studies. In contrast, other studies in England and Wales reported scope economies between water and wastewater services using a translog or a quadratic cost function, and with or without controlling for environmental variables [11, 12, 51]. Moreover, other studies by De Witte and Marques [45] found no evidence of economies of scope between water and sewerage services in the Portuguese water industry using non-parametric techniques. In contrast, other studies by Carvalho and Marques [4, 6, 9, 10] found economies of scope in the joint supply of drinking water and wastewater services in Portugal using partial or Bayesian frontier parametric techniques. Sauer [52] also found no evidence of scope economies between water and sewerage services in Germany using a McFadden variable cost function. As Saal et al. [8] and Molinos-Senante and Maziotis [11] noted, the estimation of economies or diseconomies of scope may depend on the methodology employed, i.e., parametric or non-parametric techniques; the specification of functional form, i.e., translog, quadratic; the magnitude of costs characterized in that particular industry, the choice of inputs and outputs and; the inclusion of several environmental variables.

## 4.3 Decomposition of the productivity growth

Fig 1 shows average TFP change estimations and its drivers over the years 2010–2017 for the average of Chilean water companies analyzed. It is found that on average the Chilean water and sewerage industry improved its annual productivity by 8.4% and this was attributed to both TC and SEC. In particular, TC and SEC progressed at an annual rate of 2.4% and 6.1%, respectively throughout the whole period. This finding is consistent with another study by Molinos-Senante and Maziotis [53] where the authors reported that the Chilean water industry increased its TFP by almost 10% over the period 2007–2015. Improvements in TFP for the Chilean water industry were also reported by Sala-Garrido et al. [24] and Molinos-Senante et al. [25]. In contrast, other studies by Molinos-Senante et al. [18] and Sala-Garrido et al. [19] reported a deterioration in TFP. The methodology used or the specification of the functional form of the cost function can account for differences in the estimation of TFP.

The trend in TFP and its components reveals several interesting conclusions (see Fig 1). We see that TC was progressing favorably during the years 2010–2014. This means that the

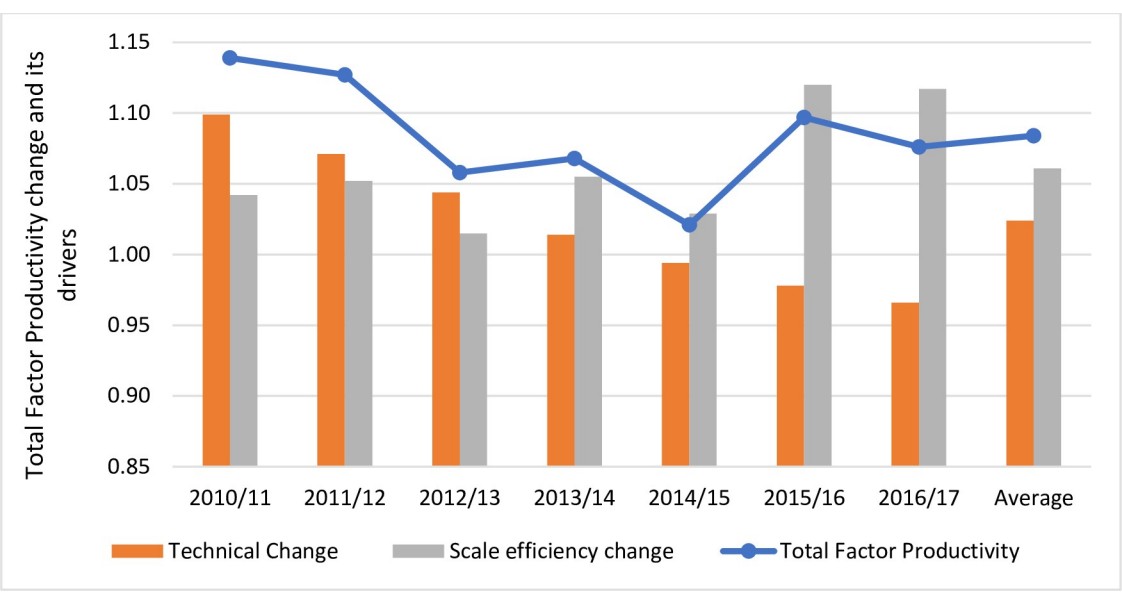

**Fig 1. Average total factor productivity change estimation and its drivers for the Chilean water companies.**

adoption of new technologies reduced average costs leading therefore, to significant improvements in TFP. We note that during that period, TC contributed more than SEC to improvements in TFP. In particular, during the years 2010/11 and 2011/12, TC increased at annual rate of almost 10% and 7%, respectively leading to considerable increases in TFP, almost 14% and 12.7% per year, respectively. However, during the years 2014/15-2016/17 TC deteriorated. In particular, the years 2016/17 the industry experienced technical regress at a rate of 3.4% per year. The negative TC was offset by the positive SEC leading therefore, to increases in productivity. Any increases in the scale of production, i.e., changes in products were higher than changes in inputs, lowered average costs leading therefore to improvements in TFP. Hence, scale effect was the major determinant of improvements in TFP; a finding which is consistent with a previous study by Molinos-Senante and Maziotis [53].

The results at an ownership level (see Fig 2) indicate that private companies (FPWCs and CWCs) performed slightly better than the public company. We found that full private and concessionary companies increased their productivity by 9.3% and 7.3% per year, respectively. Public company's TFP also improved considerably throughout the whole period at an annual rate of 7%. This result is consistent with previous studies by Molinos-Senante and Sala-Garrido [17] and Molinos-Senante et al. [18] who reported that FPWCs performed slightly better than CWCs. A study by Molinos-Senante et al. [25] showed that concessionary and private companies increased their TFP by 8.4% and 3.3% per year, respectively.

Looking at the components of the TFP, it is concluded that regardless of the ownership type, both TC and SEC contributed positively to productivity growth. However, scale effect was the major determinant of productivity. The impact of TC on full private's TFP was considerable but its impact on concessionary and public companies' TFP was negligible. The last years of our sample technical regress considerably deteriorated productivity. In particular, during the years 2016/17 full private, concessionary and public companies evidenced technical regress which was at an annual rate of 4.4%, 2.1% and 1.2%, respectively. Thus, water companies' business plans need to put more efforts on adopting best management practices to reduce overall costs than adjusting their scale of operations.

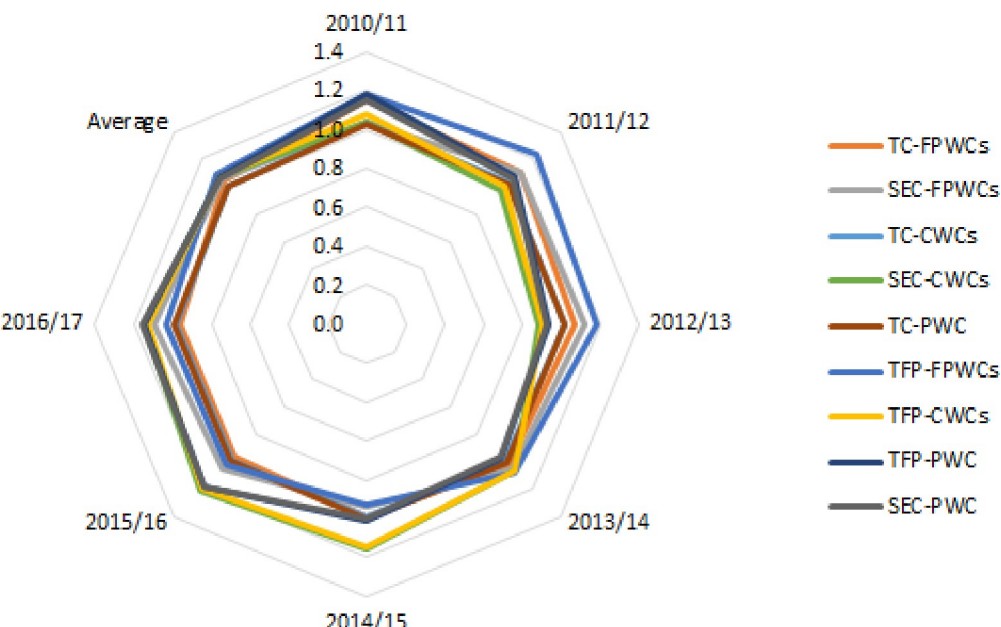

**Fig 2. Total factor productivity change estimations and its drivers (technical change -TC- and scale efficiency change -SEC-) for Chilean full private water companies (FPWCs), concessionary water companies (CWCs) and public water company (PWC).**

We now delve into the decomposition of TC into pure, non-neutral and scale-augmenting technical change (see Figs 3–5). The results indicate that PTC was the major driver of TC whereas the impact of NNTC and SATC was negligible. Pure and non-neutral TC increased at an annual rate of 2.6% and 0.1%, respectively, whereas scale-augmenting TC slightly deteriorated at a rate of 0.2% per year. Regardless of the ownership type, PTC was positive till 2013/14 whereas it deteriorated from 2014 onwards. In contrast, NNTC and SATC remained stable throughout the whole period. The positive contribution of PTC implies that the adoption of

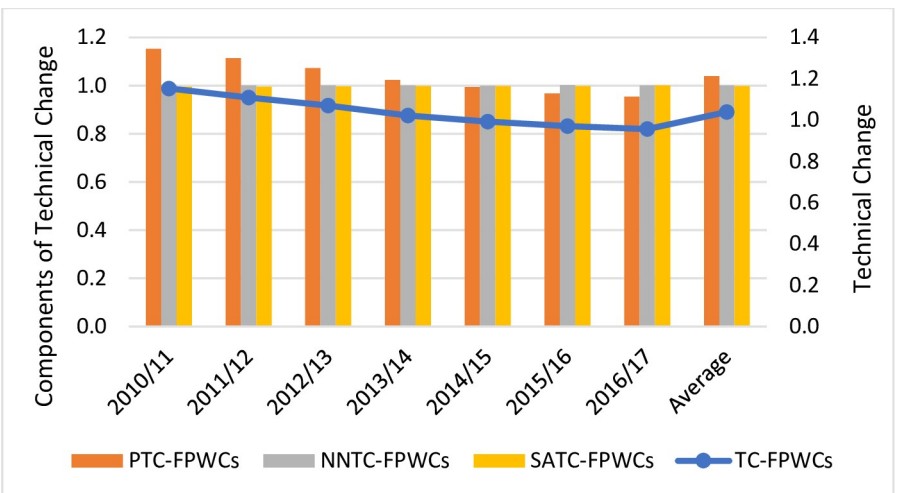

**Fig 3. Technical Change (TC) estimations and its components (pure technical change -PTC-, non-neutral technical change -NNTC and scale-augmenting efficiency change -SATC-) for Chilean full private water companies (FPWCs).**

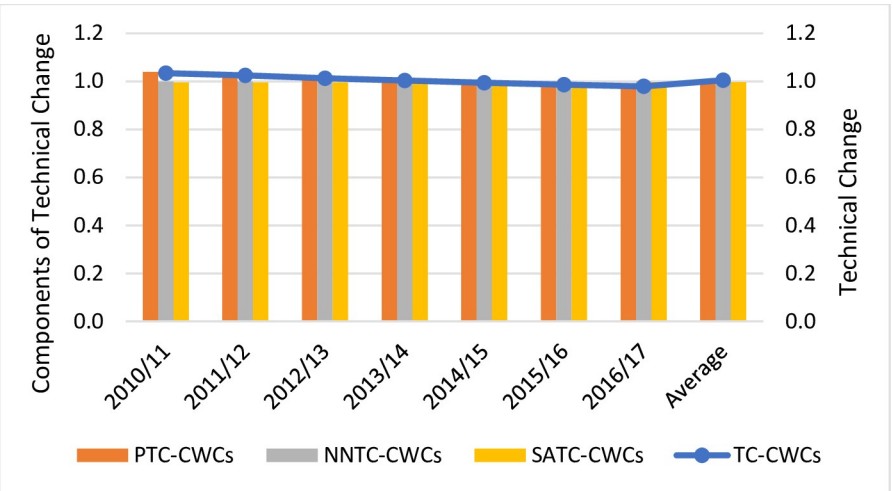

**Fig 4. Technical Change (TC) estimations and its components (pure technical change -PTC-, non-neutral technical change -NNTC and scale-augmenting efficiency change -SATC-) for Chilean concessionary water companies (CWCs).**

new technologies over time reduced overall costs. Given the price of capital and operating costs, there was a good allocation of resources as it did not favor the use of more capital than operating costs. It also appears that changes in the cost of borrowing and operating expenditure did not have a significant impact on overall costs (value of NNTC is close to unity). Finally, it seems that new technologies did not have any impact on the mix of outputs (value of SATC is close to unity).

The findings of our study can be of great interest to the policy makers for the following reasons. Firstly, they have some important implications regarding the size and configuration of the water industry. The results showed that the consolidation of the industry to increase the volume and scale of production can lead to lower costs. Moreover, the presence of integrated water and sewerage companies may not be necessary in terms of cost savings. Thus, the

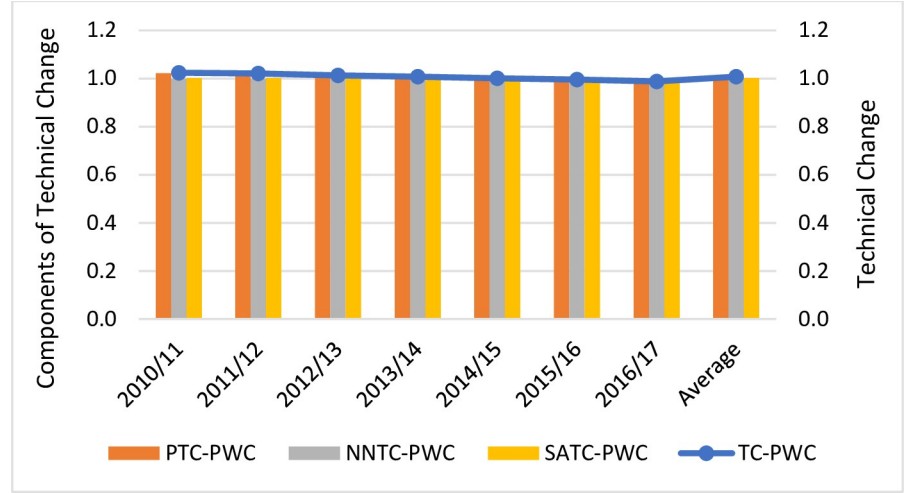

**Fig 5. Technical Change (TC) estimations and its components (pure technical change -PTC-, non-neutral technical change -NNTC and scale-augmenting efficiency change -SATC-) for Chilean public water company (PWC).**

regulator may have to promote unbundling reforms, i.e., the separation between water and sewerage services, as they can lead to lower costs for the industry. Furthermore, our methodology can aid regulated water companies to identify the determinants of productivity change which can improve performance. For instance, they can see if advances in new technology, increases in the scale of production or an efficient allocation of resources had a positive impact on productivity. The results of our study showed that adjustments in the scale of production considerably improved productivity but the companies need to put more efforts to adopt new technologies to further reduce their costs.

## 5. Conclusions

Water companies and regulators need to understand what drives costs in the water industry so that they could adopt policies and strategies to deliver water and sewerage services to customers in a sustainable and affordable manner. As part of this process, the evaluation of the cost structure of the sector and its productivity growth becomes of great importance. For instance, cost savings could be achieved by merging with other companies due to the existence of increasing economies of scale or by making technological advancements. As a result, cost savings could then be passed on to customers in terms of lower water tariffs.

In this study we used a NQCF to estimate the existence of economies of scale and scope between water and sewerage services for several water companies in Chile over the years 2010–2017. This method further allowed us to quantify the cost savings from the joint or separate production of these services. Moreover, the estimated parameters of the cost function allowed us to estimate TFP change and its determinants, TC and SEC. The first driver was further decomposed into PTC, NNTC and SATC. Results evidenced that, on average, the Chilean water companies operated under small increasing economies of scale. By contrast, negative economies of scope between water and wastewater services were reported. This implies that separating these services could reduce overall costs by 43.4% on average. The TFP results showed that the Chilean industry achieved considerable productivity gains over time. Productivity increased at a rate of 8.4% per year which was attributed to both TC and SEC. This finding means that adoption of new technologies and increases in water companies' scale of operations favored positively productivity change. Private water companies showed higher productivity growth than the public one with FPWCs performing better than concessionary ones.

Our research does not come without limitations. The study period involved was considerably small as it covered the period 2010–17. Unfortunately, due to the lack of available data for several variables such as capital expenditure we could not extend the dataset before 2010. More efforts in the future will focus on including more recent data on inputs, outputs and environmental variables and if possible collect data on additional variables that could impact water companies' performance such as volumes of sewerage collected and treated, drinking water and sewerage treatment quality.

From a policy point of view, our research could of great interest to regulators and regulated companies for the following reasons. First, we provide a methodology that allows policy makers to assess the cost structure and productive performance of the water industry. Our study showed that water companies could lower their overall costs by increasing their scale of operations. This could be done for instance through mergers with neighboring companies. Another potential source of cost savings could be the separation of water and wastewater services. Moreover, our study allows policy makers to identity how productive water companies have been over time and what drives changes in productivity. For instance, the results showed that adopting industry's best practices and adjusting scale of operations could lead to considerable

improvements in productivity. This is evident for both private and public companies shedding therefore, more evidence on the policy debate regarding ownership and productivity in water industry. More particularly, for the case of the Chilean water industry the findings could have several interesting implications. The fact that the water industry operates under increasing economies of scale means that the regulator could consider mergers among companies as a policy that could potentially lead to lower costs. Moreover, having smaller companies that offer water services only or sewerage services only could lead to lower costs. The use of new technologies could allow companies to run their business more efficiently, for instance by reducing water leakage in pipes or by using energy efficiency techniques to abstract water from different resources. This could potentially lead to higher levels of efficiency and productivity growth. Subsequently, any cost savings could pass to customers in terms of lower tariffs. An efficient operation of water and wastewater business would boost environmental sustainability contributing positively to people's health and well-being.

## Supporting information

**S1 Table. Estimated results from firm-specific dummies.**
(DOCX)

## Author Contributions

**Conceptualization:** Maria Molinos-Senante, Alexandros Maziotis.

**Data curation:** Alexandros Maziotis.

**Formal analysis:** Maria Molinos-Senante, Alexandros Maziotis.

**Funding acquisition:** Maria Molinos-Senante.

**Methodology:** Alexandros Maziotis.

**Supervision:** Maria Molinos-Senante, Alexandros Maziotis.

**Visualization:** Maria Molinos-Senante.

**Writing – original draft:** Alexandros Maziotis.

**Writing – review & editing:** Maria Molinos-Senante.

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
