## [Decision Letter · Decision Letter 0]

8 Apr 2021

PONE-D-21-05614

Productivity growth, economies of scale and scope in the water and sewerage industry: the Chilean case

PLOS ONE

Dear Dr. Molinos-Senante,

Thank you for submitting your manuscript to PLOS ONE. After careful consideration, we feel that it has merit but does not fully meet PLOS ONE’s publication criteria as it currently stands. Therefore, we invite you to submit a revised version of the manuscript that addresses the points raised during the review process.

We look forward to receiving your revised manuscript.

Kind regards,

Bing Xue, Ph.D.

Academic Editor

PLOS ONE

Journal Requirements:

Reviewers' comments:

Reviewer's Responses to Questions

**Comments to the Author**

1. Is the manuscript technically sound, and do the data support the conclusions?

Reviewer #1: Yes

Reviewer #2: Partly

2. Has the statistical analysis been performed appropriately and rigorously? 

Reviewer #1: Yes

Reviewer #2: Yes

3. Have the authors made all data underlying the findings in their manuscript fully available?

Reviewer #1: Yes

Reviewer #2: No

4. Is the manuscript presented in an intelligible fashion and written in standard English?

Reviewer #1: Yes

Reviewer #2: Yes

5. Review Comments to the Author

Reviewer #1: Molinos-Senante and Maziotis present a case study for Chilean water utilities in a very good manner. I must also say that I have been following their work for a while and I think Maria is doing great work on water utilities. The manuscript describes the use of a quadratic function for costs using actual data. Conclusions would have an impact on public policy, and results and methods are sound. However, I have a couple of comments regarding the manuscript structure and some topics that to discuss more profoundly:

I. The Introduction gives some background regarding economies of scale and scope. I think the authors should also discuss how both concepts are related to regulation and water supplies. Moreover, authors could discuss here, in another piece, how climate and climate change affect pricing. I would also suggest introducing and discussing whether the Chilean pricing procedures (against the "model company") and parametrization affect cost functions.

II. Lines 101-128 are more a discussion (or literature review) than a methodological description. I would like to suggest a critical appraisal of the quadratic function as compared to other approaches (just a couple of lines touched this issue).

III. To help readiness, I suggest adding a table of variables. Also, please provide more details regarding the statistical significance test.

IV. In lines 277-278 the authors report the importance of water sources. Please, briefly discuss the importance of groundwater due to availability, depth, and operational costs.

V. In lines 295-297, please elaborate on the importance of economies of scale in Chile.

VI. Do regulatory strength influence the economies of scale and scope? (lines 298-325)

VII. What did drive TFP changes in 2010/11, 2011/12, 2014/15, and 2016/17? (lines 345-356)

VIII. I am not a native English speaker, but there few grammar and word choices that authors should correct. (e.g. lines 268, 252)

IX. Please, explore other visualisation options for figures 2 and 3.

Reviewer #2: This study beholds practical values as examining the existence of economies of scale and scope in the Chilean water and sewerage industry. The writing is fluent, plain, and readable. However, in my view, there still exist some places for improvement:

First of all, certain aspects of contents in this study need to be expanded to increase its theoretical contributions. As the introduction part lacks address the value (or the aim) of examining the existence of economies of scale and scope: will confirming this existence help to industry regulation, industry investment, or enhancement of citizen’s well-being? And how? I do not see adequate background knowledge on that.

For another thing, focusing on directly economies of scale in the Chilean water and sewerage industry could easily lose reading interests from international readers out of Chile. Thus this study should offer more content under the theme as “What can other countries learn from Chile’s case.” I see such attempts in Section 4: the attempts were good; however, those discussions were not conducted in-depth. For instance, from Line 304, Page 16 to Line 325, Page 17, the authors compared their findings with previous studies in some other countries; however, the comparison were limited to simply the results (existence of economies of scale or not) and lacked at providing some possible explanations to that divergence in results. Although the authors quoted Saal et al. (2013) and Molinos-Senante et al. (2017) acclaiming that the result divergence was caused by the methodology used, I am more curious about: (1) What is the authors’ opinion about this divergence in results? Do you believe in the methodology-caused explanation? (2) If yes, considering the capture of economies of scale is method-specific, can you offer some more comments on the previous methods, conduct comparisons among those methods, and then acclaim the advantage of the current method in capturing economies of scale based on the results? (3) Aside from the methodology-caused explanation, are there any other explanations for the result divergence? More specifically, as is mentioned in the introduction part, this study is among the earliest studies examining economies of scale and scope in developing countries; thus, could more explanations be offered from the perspective of the developing/developed country difference?

As for the methodology, considering the model this study relied on upon to estimate is very classic, sensitivity analysis and heterogeneity analysis should be considered to add the estimation validity and offer more research implications.

Besides, I noticed that data was collected for the period 2010-2017 (which a relatively small-size sample); the authors should address why data before that time range is not accessible and write it into the research limitation.

Finally, more details about the case of Chile’s water and sewerage industry system and development are expected to be offered to readers as a research background. Correspondingly, practical implications in the conclusion part (such as content from Line 430, Page 21 to the end) should be discussed more related to Chile’s situations.

6. PLOS authors have the option to publish the peer review history of their article (what does this mean?). If published, this will include your full peer review and any attached files.

Reviewer #1: **Yes: **Diego Rivera

Reviewer #2: No

---

## [Author Response · Author response to Decision Letter 0]

20 Apr 2021

See document attached "Response to Reviewers"

---

## [Decision Letter · Decision Letter 1]

5 May 2021

Productivity growth, economies of scale and scope in the water and sewerage industry: the Chilean case

PONE-D-21-05614R1

Dear Dr. Molinos-Senante,

We’re pleased to inform you that your manuscript has been judged scientifically suitable for publication and will be formally accepted for publication once it meets all outstanding technical requirements.

Kind regards,

Bing Xue, Ph.D.

Academic Editor

PLOS ONE

Additional Editor Comments (optional):

Reviewers' comments:

Reviewer's Responses to Questions

**Comments to the Author**

1. If the authors have adequately addressed your comments raised in a previous round of review and you feel that this manuscript is now acceptable for publication, you may indicate that here to bypass the “Comments to the Author” section, enter your conflict of interest statement in the “Confidential to Editor” section, and submit your "Accept" recommendation.

Reviewer #1: All comments have been addressed

Reviewer #2: All comments have been addressed

2. Is the manuscript technically sound, and do the data support the conclusions?

Reviewer #1: Yes

Reviewer #2: Yes

3. Has the statistical analysis been performed appropriately and rigorously? 

Reviewer #1: Yes

Reviewer #2: Yes

4. Have the authors made all data underlying the findings in their manuscript fully available?

Reviewer #1: Yes

Reviewer #2: Yes

5. Is the manuscript presented in an intelligible fashion and written in standard English?

Reviewer #1: Yes

Reviewer #2: Yes

6. Review Comments to the Author

Reviewer #1: (No Response)

Reviewer #2: (No Response)

7. PLOS authors have the option to publish the peer review history of their article (what does this mean?). If published, this will include your full peer review and any attached files.

Reviewer #1: **Yes: **DIEGO RIVERA-SALAZAR

Reviewer #2: No

---

## [Editor Report · Acceptance letter]

11 May 2021

PONE-D-21-05614R1 

Productivity growth, economies of scale and scope in the water and sewerage industry: the Chilean case 

Dear Dr. Molinos-Senante:

I'm pleased to inform you that your manuscript has been deemed suitable for publication in PLOS ONE. Congratulations! Your manuscript is now with our production department. 

Kind regards, 

on behalf of

Professor Bing Xue 

Academic Editor

PLOS ONE